# The LIM Domain Protein BmFHL2 Inhibits Egg Production in Female Silkworm, *Bombyx mori*

**DOI:** 10.3390/cells12030452

**Published:** 2023-01-31

**Authors:** Qian Yuan, Xiaoning Sun, Riming Lu, Zhigang Qu, Xueyan Ding, Taiming Dai, Jianfeng Qiu, Yumei Tan, Ruihong Zhu, Zhonghua Pan, Shiqing Xu, Yanghu Sima

**Affiliations:** 1School of Biology and Basic Medical Sciences, Suzhou Medical College, Soochow University, Suzhou 215123, China; 2Institute of Agricultural Biotechnology & Ecology (IABE), Soochow University, Suzhou 215123, China

**Keywords:** BmFHL2, *Bombyx mori*, egg formation, 30K protein

## Abstract

The female *Bombyx mori* accumulates a large amount of egg proteins, mainly Vg and 30K, during egg formation to provide nutrition for embryo development. The synthesis and transport of Vg have been extensively studied, particularly the regulation of Vg transcription induced by 20E; however, the mechanism of 30K protein synthesis is poorly studied. As a model organism of the order Lepidoptera, *B. mori* has high reproduction potential. In the present study, we found that the FHL2 homologous gene (*BmFhl2*) in *B. mori* is involved in inhibiting female egg formation by influencing the synthesis of 30K protein. Interference of *BmFhl2* expression in silkworm females increased 30K protein synthesis, accelerated ovarian development, and significantly increased the number of eggs produced and laid; however, the 20E pathway was inhibited. The transcription levels of *Vg* and *30Kc19* were significantly downregulated following *BmFhl2* overexpression in the silkworm ovarian cell line BmN. The Co-IP assay showed that the potential binding protein of BmFHL2 included three types of 30K proteins (30Kc12, 30Kc19, and 30Kc21). These results indicate that BmFHL2 participates in egg formation by affecting 30K protein in female *B. mori*.

## 1. Introduction

The protein FHL2 belongs to the FHL (four-and-a-half LIM domain) protein family and contains four semi-LIM domains (half LIM domains at the N-terminus), with each LIM domain containing two zinc finger structures [1,2]. Unlike the DNA binding functions of many zinc finger structures, current studies in mice and humans have revealed that the LIM domain is involved only in protein–protein interactions, thus regulating different physiological processes [3]. FHL2 is expressed in a variety of tissues, serving as a transcriptional regulator through its interaction with various transcription factors [4], directing the functions of the protein complexes formed [5]. In mammals, several studies have shown that FHL2 is involved in many reproductive processes, including the regulation of female follicle development [6], ovulation, and ovarian development [4]; however, there are no research reports on the effect of FHL2 regulation on insect female reproduction.

Compared to mammals, insects have a unique ovarian structure and a specific egg development regulation mechanism. Insect ovaries can be divided into panoistic and meroistic types based on whether they contain trophoblasts [7]. *B. mori* ovaries belong to the meroistic type. Trophoblasts in the pupal stage transport nutrients through the oocytes, which promote the growth of primary oocytes, and following the exhaustion of nutrients from trophoblasts, follicular cells act as nutrient-producing cells [8,9]. Primary oocytes complete their growth after the formation of the yolk membrane (which later becomes the eggshell) [10], and 20E (20-hydroxyecdysone) can promote the maturation of oocytes [11].

The nutrients required for the development of silkworm embryos are mainly derived from yolk proteins, i.e., 40% Vn (vitellin), 35% 30K (small molecule lipoprotein), and 25% ESP (egg-specific protein) [12]. Vg (a precursor of Vn) is transcribed in the fat body from the pupation stage of silkworms, and its content in the pupae gradually increases in the early stage and rises sharply in the middle and late stages [13]. This protein is transported to the ovary through hemolymph by VgR (Vg receptor)-mediated transport [14]. The 30K protein is a typical secretory protein synthesized in the fat body and then secreted into the hemolymph; it is then transported to the egg and further degraded in the embryonic intestinal lumen [15,16]. The 30Kc19 protein is a member of the 30K protein family (30Kc6, 30Kc12, 30Kc19, 30Kc21, and 30Kc23), and it is the most abundant 30K protein [17]. 30Kc19 is also a cell-penetrating protein [18]. Several studies have demonstrated that Vg synthesis in silkworms is mainly regulated by 20E, which activates the downstream signaling pathways by binding to USP and EcR receptors and promotes the expression of transcription factors such as E75, E74, POUM2, and BrC, thereby promoting the transcription of Vg [10,11,19]. The synthesis of the 30K protein is inhibited by JH (juvenile hormone) [20,21]. The embryonic development of silkworm is different from mammalian embryonic development (nutrients are derived directly from the mother), and its yolk is synthesized by fat bodies and deposited as the main nutrient into the developing oocytes. In the present study, the effect of FHL2 on the reproduction of female insects was studied for the first time by using silkworms as the test material. We confirmed the close association between BmFHL2 and 30K protein by RNA interference in individuals, overexpression in BmN cells, and Co-IP assay. The study provides a theoretical basis for exploring the regulatory mechanism of silkworm egg formation.

## 2. Materials and Methods

### 2.1. Experimental Animals and Cells

The classic genetic strain of silkworm, P50 (Dazao), was used in the present study and maintained in the School of Biology and Basic Medical Sciences, Medical College of Soochow University. The larvae of silkworm were reared on fresh mulberry leaves under 12 h light (66 lux)/dark conditions at 25 °C ± 1 °C with 65% ± 5% humidity. We chose larvae of similar weight and separated male and female individuals at the beginning of the fifth instar. The tissue materials, including ovary, testis, fat bodies, hemolymph, and midgut, were collected at the 5th instar 3 days (5L3), 5th instar 6 days (5L6), wandering stage (W), pupal stage 0 day (P0), pupal stage 1 day (P1), pupal stage 3 days (P3), and pupal stage 6 days (P6). The silkworm was dissected on the dissection plate, and the extracted tissues were washed with DEPC and placed in the centrifuge tube, stored at −80 °C. All samples were collected and mixed for three silkworm populations. After disinfection with 75% ethanol, the epidermis was punctured, and hemolymph was collected using an Eppendorf tube in an ice bath. Equal volumes of hemolymph from 3 individuals were mixed to form a hemolymph sample. Hemocytes were collected after centrifugation at 3000 rpm for 10 min at 4 °C and used for RNA extraction. The silkworm ovary cell line BmN was used for constant temperature culture at 26 °C. The cells were cultured in a complete medium containing 12% fetal bovine serum (Biological Industries, Beit Haemek, Israel) and 88% Grace medium (GIBCO, Carlsbad, CA, USA).

### 2.2. Evolutionary Tree Analysis

The SMART protein structure prediction website (http://smart.embl-heidelberg.de/, accessed on 1 January 2020.) was used to predict protein domains. By using NCBI BLAST for the amino acid sequence of FHL2 in 14 other species, a phylogenetic tree was constructed using the software MEGA7. The parameters were set as molecular phylogenetic analysis with 1000 times bootstrap.

### 2.3. Gene Expression Analysis

Total RNA was extracted from cells or tissues using TRIzol^TM^ reagent (Thermo Fisher, Waltham, MA, USA). The cDNA was synthesized using a PrimeScript^TM^ RT reagent kit with a gDNA eraser (TaKaRa, Dalian, China) in accordance with the manufacturer’s instructions. TB Green^®^ Fast qPCR Mix (TaKaRa, Dalian, China) was configured in a 20 μL reaction system for real-time quantitative fluorescent polymerase chain reaction (qPCR). *B. mori Rp49* (ribosomal protein 49) was used as a positive control.

### 2.4. RNA Interference

A small interfering RNA (siRNA) of the *BmFhl2* gene (gene ID: 101741761) was designed and synthesized by GenePharma (Shanghai, China). Primer sequences (siRNA—*BmFhl2* 405 and siRNA—*BmFhl2* 777) are shown in Appendix A. At 3 h after pupation, 10 μg siRNA was injected (siRNA—*BmFhl2* 405:siRNA—*BmFhl2* 777 = 1:1), and the same amount of scrambled siRNA (sequences are shown in Appendix A) was used as a negative control (si NC). The ovaries and fat bodies were collected at 24 and 48 h after injection, and the efficiency of interference was determined by qPCR.

### 2.5. Egg Count Statistics

After gene knockdown, group mating was performed for 6 h as follows: female si *BmFhl2* × male si NC; Female si NC × male si NC. The number of eggs laid, and the number of eggs left in the abdomen were counted at 16 h after mating. The calculation formula was as follows:number of eggs produced = number of eggs laid + number of eggs left in the abdomen
egg laying rate = number of eggs laid/number of eggs produced

### 2.6. Western Blotting (WB)

Protein extraction was performed with 100 μL RIPA lysis buffer (Beyotime, Shanghai, China) containing 1% phenylmethylsulfonyl fluoride (PMSF, Beyotime, Shanghai, China) by using the TGX Stain-Free FastCast Acrylamide Kit (Bio-Rad, 1610182, Hercules, CA, USA). The extracted proteins were separated by sodium lauryl sulfate-polyacrylamide gel electrophoresis (SDS-PAGE) and then transferred to a semidry transfer membrane. The membrane was sealed with a sealing solution (Bio-Rad, P0023B, Shanghai, China) at 25 °C for 2 h and then incubated with the following antibodies at 4 °C for 12 h: purified anti-Vg antibody (1:1000), anti-30Kc19 antibody (1:1000), anti-V5 tag antibody (1:1000, AB_2533339, Thermo Fisher, Waltham, MA, USA), and anti-α-tubulin antibody (1:5000, T0033, Affinity). After three times washing with Tris-buffered saline (TBS) containing 0.05% Tween 20 (TBST; pH 7.5), the membrane was incubated with HRP-conjugated anti-mouse IgG (1:5000, Bioworld Technology, St Louis Park, MN, USA) at 37 °C for 2 h. After washing the membrane again with TBST three times, an appropriate amount of ECL (Bio-Rad) coloring solution (1:1) was added, and the membrane was kept incubated in dark. Image Lab software was used to process and analyze the generated images.

### 2.7. Construction of Overexpressing Plasmids

By using the gonadal cDNA of female silkworm 5L3 as the template, the *BmFhl2-X6* CDS sequence was amplified by high-fidelity PCR (KOD-201, Toyobo, Shanghai, China). The primer sequence is shown in Appendix A. The PCR product was recovered, attached to the pMD-19T carrier (D102A, TaKaRa, Dalian, China), and extracted after conversion. FastDigest *Kpn*I (FD0524, Invitrogen, Vilnius, Lithuania) and *Not*I (FD0595, Invitrogen, Vilnius, Lithuania) double-digested pIZT empty vectors and pMD-19T vectors were successfully constructed. T4 DNA ligase (2011A, TaKaRa, Dalian, China) was used to link the digestion products. Recombinant plasmids pIZT/*BmFhl2-X2* and pIZT/*BmFhl2-X6* were sequenced by Sangon Biotech Co., Ltd. (Shanghai, China). After the sequence and target fragment were corrected, the recombinant plasmid was extracted with a large extraction kit (12362, Qiagen, Hilden, Germany).

### 2.8. Transfection

The plasmid medium was prepared by adding 2500 ng plasmid to 200 μL of serum-free medium. A plasmid equivalent volume of transfection reagent (X-tremeGENE^TM^ HP DNA Transfection Reagent, 6366236001, Roche, Mannheim, Germany) was added to another 200 μL of serum-free medium to prepare the transfection reagent medium. The plasmid medium was added to the transfection reagent medium, gently mixed, and incubated for 30 min to form the transfection complex. The old culture medium was completely removed from the 6-well plate, the cells were rinsed three times with the serum-free culture medium, and finally 1.5 mL of serum-free culture medium was added to each well. After incubation, 400 μL of the transfection complex was added to each well. After incubation for 6–8 h, the cell state was observed; the medium was replaced with a complete medium; the cells were collected at 24, 48, and 72 h in culture, and qPCR was performed.

### 2.9. Co-Immunoprecipitation (Co-IP) and Mass Spectrometry

The transfected cells in 6-well plates were washed twice with an appropriate amount of PBS. Next, 100 μL protein lysate, 1 μL PMSF, and 1 μL phosphatase inhibitor (PhosSTOP, Sigma-Aldrich, St. Louis, MO, USA) were added to each well. The protein extract was added to the 6-well plate. The cells were scraped off with a cell scraper, mixed, and transferred to a centrifuge tube; the cells were then subjected to mixing at 4 °C for 30 min. After centrifugation at 12,000 rpm at 4 °C for 30 min, the supernatant was transferred to a new centrifuge tube. Subsequently, 1 μL LPMSF, 1 μL phosphatase inhibitor, and 1.5 μL anti-V5 tag antibody (AB_2533339, Thermo Fisher, Waltham, MA, USA) were added to each tube. The tubes were subjected to gentle shaking at 4 °C overnight. The supernatant was discarded, and 35 μL of fully suspended ProteinA+G Agarose (CW0349S, CWBIO, Beijing, China) was added to the centrifuge tube. Next, 1 mL of pre-cooled PBS was added and mixed. The mixture was then centrifuged at 4 °C at 3000 rpm for 2 min. The supernatant was discarded, and the cleaning process was repeated 3 times. The washed ProteinA+G Agarose was added to the mixture for overnight incubation in a homogenizing system at 4 °C for 2 h. The mixture was then centrifuged at 4 °C at 3000 rpm for 30 s, and the supernatant was discarded. Next, 500 μL PBS was added, mixed gently, and centrifuged at 3000 rpm at 4 °C for 30 s; the supernatant was discarded, and the process was repeated 3 times. The final precipitate was added to 40 μL 2× SDS-PAGE loading buffer, boiled at 99 °C for 10 min, and centrifuged at 12,000 rpm at 4 °C for 5 min. The supernatant was taken and subjected to silver staining. The differential bands were excised and sent to Shanghai Zhongke New Life Company for mass spectrometry analysis. The polypeptide molecules were identified and analyzed after processing with the HCD (higher-energy collision dissociation) method.

### 2.10. Statistical Analysis

Image-Pro Plus v6.0 and GraphPad Prism v8 (GraphPad, San Diego, CA, USA) were used for image and data processing, respectively. Multiple *t*-test (one per row) analysis was performed using the Holm–Sidak method.

## 3. Results

### 3.1. Phylogenetic Tree Construction and Analysis of the Spatiotemporal Expression of BmFhl2

Silkworm *BmFhl2* contains six different shear bodies (Appendix A). Phylogenetic analysis of BmFHL2-X2 showed that the FHL2 of silkworm was closely related to that of *Manduca sexta*, *Amyelois transitella* (Figure 1A), and *Spodoptera litura*, *Spodoptera frugiperda*, *Helicoverpa armigera*, and *Trichoplusia ni*, members of the order Lepidoptera, which are distant from fruit flies, *Homo sapiens*, and mice. FHL2 is suggested to be relatively conserved during the evolution of the order Lepidoptera and is predicted to perform similar functions.

The present study investigated the changes in the expression levels of *BmFhl2* from the fifth instar to the early stage of pupa. RT-PCR results showed that the expression level of *BmFhl2* in ovary and fat body first increased, then decreased, and then again increased from the mid-fifth instar to the wandering stage to the early stage of pupa, with apparent peaks and troughs in the expression levels. The expression of *BmFhl2* decreased after the pupation stage (Figure 1B), and a similar expression pattern was noted in the testis, hemolymph, and midgut (Appendix A). On the basis of this expression pattern, it is speculated that BmFHL2 may participate in the reproduction process of female silkworms.

### 3.2. Knockdown of BmFhl2 Promotes Egg Production of Female Silkworm

To determine the effect of *BmFhl2* on female reproduction in the silkworm pupal stage, RNA interference (RNAi) was performed on *BmFhl2* of female pupa at 3 h after pupation. Ovaries and fat bodies were collected at 24 and 48 h for interference confirmation, and the interference efficiency reached >33.3% at 48 h (Figure 2A,B). The ovarian development in the RNAi group was faster than that in the control group at 48 h (Figure 2C). The results of the egg laying assessment showed that after the knockdown of *BmFhl2* in female silkworms, the number of eggs produced by single moths increased by an average of 40, and the number of eggs laid by single moths increased by an average of 60 (Figure 2D,E); moreover, no significant difference was observed in the egg laying rate (Figure 2E). The results confirmed that knockdown of *BmFhl2* in female silkworms promoted egg formation.

### 3.3. Knockdown of BmFhl2 Promotes the Synthesis of 30Kc19 Protein

The proteins of silkworm eggs mainly consist of Vg and 30K. qPCR assay was performed to determine the transcription levels of *30Kc19* and *Vg* in the fat body. Compared to that in the control group, the expression levels of *Vg* and *30Kc19* were significantly upregulated after *BmFhl2* knockdown, while the expression level of *VgR* was significantly upregulated in the fat body (Figure 3A). In addition, the results of Western blotting assay showed that the expression levels of 30Kc19 and Vg proteins increased after the knockdown of *BmFhl2* (Figure 3B). The expression of 30Kc19 significantly increased, but that of Vg showed no significant difference (Figure 3C). The analysis of the transcription level of the related receptor genes downstream of 20E showed that after *BmFhl2* knockdown, the levels of both *EcR* and *USP* were significantly decreased at 24 h; however, no significant difference was observed in the expression level of E75a (Figure 3D). These results showed that *BmFhl2* knockdown can promote the transcription of *Vg* and *30Kc19* in the fat body and increase the expression of the 30Kc19 protein; however, it did not promote the synthesis of the egg protein through the activation of the 20E pathway.

### 3.4. BmFhl2 Overexpression in BmN Cells Inhibits the Transcription of 30Kc19 and Vg

According to the structure of six different BmFhl2 splices, BmFhl2-X2 and BmFhl2-X6 binding proteins were predicted to be more abundant (Appendix A). Therefore, *BmFhl2-X2* and *BmFhl2-X6* were selected to construct the overexpression vector (Appendix A), and this vector was overexpressed in BmN cells (Figure 4A). The overexpression efficiency was determined by qPCR and WB (Figure 4B,C). After overexpression of the two splices of *BmFhl2*, the expression levels of *Vg*, *30Kc19*, and *VgR* were significantly downregulated at 24, 48, and 72 h after transfection (Figure 4D,F). These results suggest that *BmFhl2* can inhibit the transcription of *30Kc19* and *Vg*.

### 3.5. BmFHL2 May Bind to 30K Protein to Regulate Egg Formation

To further investigate the function of BmFHL2, differential bands were selected by silver staining of bands in the co-immunoprecipitation (Co-IP) assay (Figure 5A), and reproduction-related proteins, including 30Kc19, 30Kc12, and 30Kc21, were screened by mass spectrometry (Appendix A and Appendix A). Compared to that in the control group, the expression levels of all three 30K proteins were significantly downregulated at the transcription level (Figure 4D and Figure 5B,C); thus, it is speculated that BmFHL2 may directly or indirectly bind to the 30K protein to participate in female egg formation.

## 4. Discussion

In the present study, the function of the FHL2 protein in reproduction was characterized in female *B. mori*. BmFHL2 inhibited the transcription of *30Kc19* and *Vg* in the fat body of silkworms, which reduced the synthesis of 30K protein and consequently reduced the amount of egg produced. *BmFhl2* was also highly expressed in the testis, hemolymph, and midgut of the silkworm (Appendix A). It is thought that this gene has multiple functions, for example, in mammals, the FHL2 protein regulates early sex differentiation and testicular formation and development [22,23,24].

The synthesis of egg protein in insects is mainly regulated by endocrine systems. Estradiol injection or feeding increases Vg synthesis in silkworms [25]. Previous studies have shown that JH regulates vitellogenesis and oogenesis in *Cimex lectularius* and *Aedes aegypti* mosquitoes [26,27]. In *Haemaphysalis longicornis* and brown planthopper, the TOR pathway is also involved in the synthesis of Vg [28,29]. In conclusion, JH and 20E do not have the same regulatory effect on egg protein in different insects. Several studies on silkworms mainly focused on the regulatory effect of 20E on Vg transcription and transport [13,19,30]; moreover, although 30K synthesis in the fat body was inhibited by JH [21], the specific regulatory mechanism remains unclear. The results of the present study showed that *BmFhl2* knockdown significantly upregulated the expression of *Vg* and *30Kc19* at the transcription level (Figure 4A,C); however, the expression levels of the 20E downstream receptor genes *EcR* and *USP* were significantly downregulated in fat body at 24 h after knockdown (Figure 4D). On the basis of this finding, we speculated that the egg-forming capacity enhanced by knockdown of *BmFHL2* was not regulated by the 20E pathway.

The proteins pulled out by Co-IP in the present study included glutamate dehydrogenase (Appendix A). Glutamate dehydrogenase, located in the mitochondria of cells, can catalyze the conversion of glutamate into α-ketoglutaric acid and NH^4+^, regulate amino acid-induced insulin secretion, and induce hyperinsulinemia after mutation [31]. Insulin-related peptide signaling pathways regulate oogenesis in insects. Previous studies have shown that insulin signaling in *Drosophila melanogaster* and *Caenorhabditis elegans* directly regulates oocyte growth and maturation [32,33]. Insulin-like growth factors in silkworms can regulate the development of ovaries [34]. In addition, in the present study, interference with female silkworm *BmFhl2* not only increased the number of eggs produced and laid but also promoted the development of ovaries, and ovarian development was significantly accelerated as compared to that in the control group at 48 h after the interference (Figure 2C). FHL2 deficiency improves insulin secretion from beta cells and improves glucose tolerance in mice [35], and co-immunoprecipitation studies confirmed that interaction between IGFBP-5 (insulin-like growth factor binding protein 5) and FHL2 occurs in whole cells [36]. It is thought that BmFHL2 may affect the development of silkworm ovaries through the insulin signaling pathway and ultimately affect the formation of eggs; however, the specific regulatory mechanism of this phenomenon needs to be further elucidated. As a component of silkworm yolk protein, the 30K protein not only provides nutrients for embryonic development [37], but also participates in inhibiting programmed cell death and antifungal cellular immunity [38,39,40]. Thus, 30K may also be an important factor that affects ovarian development. According to existing studies, FHL2 is an adaptor protein that may simultaneously bind different proteins [5], so we hypothesize that BmFHL2 may act as a transcriptional regulator to regulate the transcription of 30K protein while binding to 30K protein to participate in egg formation in *B. mori*.

## 5. Conclusions

BmFHL2 is an evolutionarily conserved protein in lepidopteran insects. It participates in silkworm egg formation by influencing the synthesis of 30K protein or binding (directly or indirectly) with 30K protein, but the exact mechanism remains unclear. Further study of the regulatory mechanism of BmFHL2 on 30K protein will provide important theoretical bases for the reproduction in insects.

## Figures and Tables

**Figure 1 cells-12-00452-f001:**
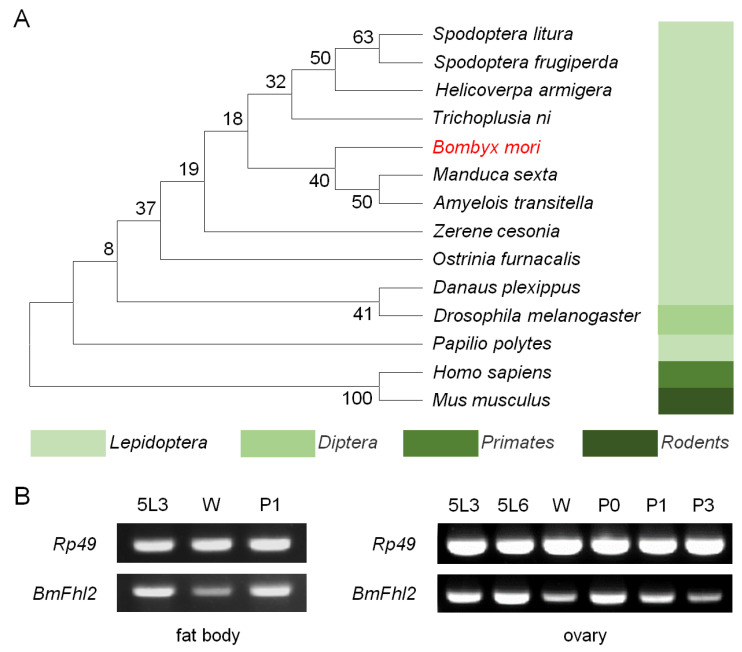
Bioinformatics analysis of BmFHL2. (**A**) Phylogenetic tree of BmFHL2. The amino acid sequence of BmFHL2-X2 was used for phylogenetic analysis. On the basis of the amino acid sequence of the BmFHL2 protein, NCBI Blast was performed with homologous sequences of the other species, and MEGA7 was used to construct a phylogenetic tree. The red mark indicates *B. mori*. The comparison species involve four orders among Lepidoptera, Diptera, Rodentia, and primates, including 14 species in total. (**B**) Expression levels of *BmFhl2* in fat body and ovary at different time points. Fat bodies and ovaries were collected at different time points, and total RNA was extracted for RT-PCR analysis. *Rp49* was used as a reference gene. All samples were collected and mixed for three silkworm populations. 5L3: 5th instar 3 days, 5L6: 5th instar 6 days, W: wandering stage, P0: pupal stage 0 days, P1: pupal stage 1 day, P3: pupal stage 3 days.

**Figure 2 cells-12-00452-f002:**
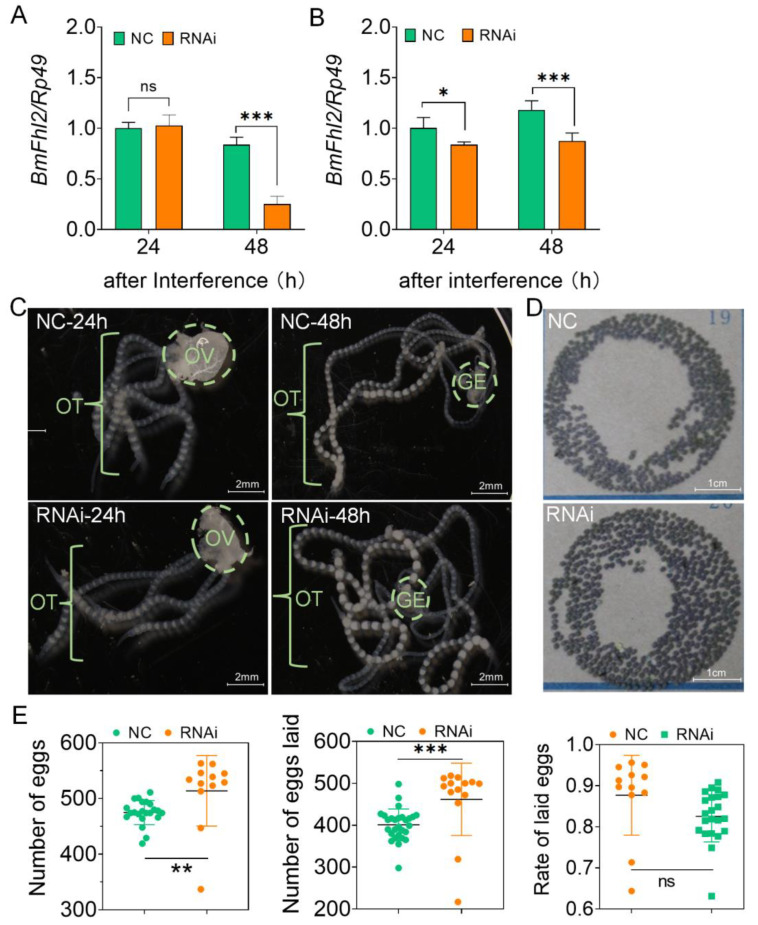
Effects of *BmFhl2* knockdown on the reproduction of female silkworms. (**A**) Efficiency of RNAi for *BmFhl2* expression in the ovary. Total RNA of the ovaries was extracted for qPCR analysis at 24 and 48 h after RNAi. All samples were collected and mixed for three silkworm populations. (**B**) Efficiency of RNAi for *BmFhl2* expression in the fat body. Total RNA of the fat bodies was extracted for qPCR analysis at 24 and 48 h after RNAi. All samples were collected and mixed for three silkworm populations. (**C**) Ovarian development morphology after *BmFhl2* knockdown. OT: ovarian tube, OV: ovary, GE: germarium. (**D**) Oviposition map of the NC and RNAi groups after *BmFhl2* knockdown. The female moths of the two groups were mated with NC male moths. (**E**) Investigation on egg production, number of eggs laid, and the egg laying rate after RNAi. n = 11–28. Mean ± SD, N = 3. *, *p* ≤ 0.05; **, *p* ≤ 0.01; ***, *p* ≤ 0.001; ns: no significant difference. *Rp49* was used as a reference gene. NC: injection of NC siRNA; RNAi: injection of *BmFhl2* siRNA.

**Figure 3 cells-12-00452-f003:**
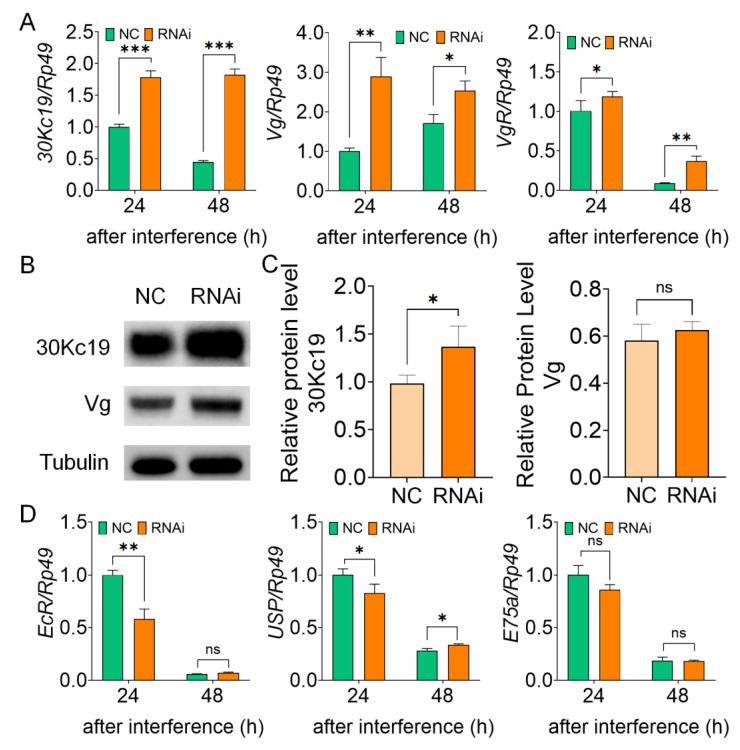
Effects of *BmFhl2* knockdown on the expression of egg protein and the 20E receptor gene. (**A**) qPCR results of the expression levels of *30Kc19*, *Vg*, and *VgR* in fat body at 24 and 48 h after interference. (**B**) WB results of the expression levels of *30Kc19* and *Vg* in fat body at 48 h after interference. (**C**) Quantitative analysis of WB results. (**D**) qPCR results of the expression of *EcR*, *USP*, and *E75a* in fat body at 24 and 48 h after interference. *, *p* ≤ 0.05; **, *p* ≤ 0.01; ***, *p* ≤ 0.001; ns: nonsignificant difference. *Rp49* was used as the qPCR reference gene, while tubulin was used as the WB reference protein. Mean ± SD, N = 3. All samples were collected and mixed for three silkworm populations.

**Figure 4 cells-12-00452-f004:**
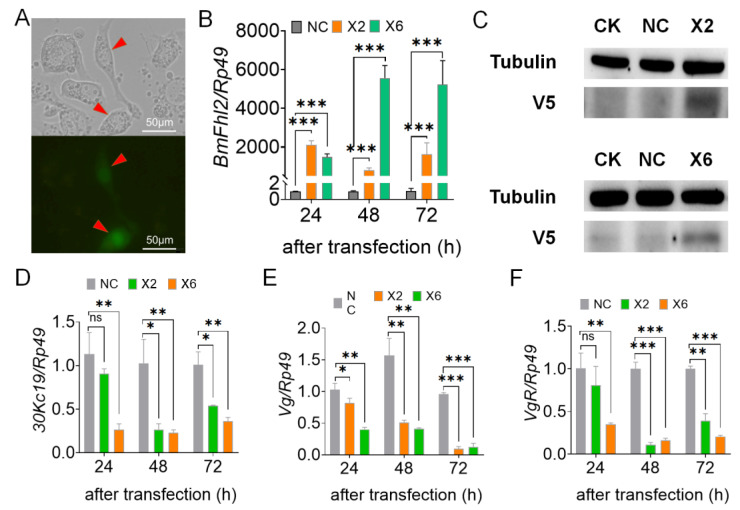
Effects of *BmFhl2* overexpression on the egg protein-related genes. (**A**) GFP fluorescence assay after overexpression. Transgenic cells under normal light and stimulated light are shown. The red arrows indicate successfully transfected cells. (**B**) Confirmation of *BmFhl2-X2* and *BmFhl2-X6* overexpression in BmN cells at the transcriptional level. (**C**) WB assay was performed at 48 h after *BmFhl2* overexpression. (**D**–**F**) qPCR analysis of the expression levels of *30Kc19*, *Vg*, and *VgR* at 24, 48, and 72 h after overexpression. *, *p* ≤ 0.05; **, *p* ≤ 0.01; ***, *p* ≤ 0.001; ns: nonsignificant difference. *Rp49* was used as the qPCR reference gene, while tubulin was used as the WB reference protein. Mean ± SD, N = 3.

**Figure 5 cells-12-00452-f005:**
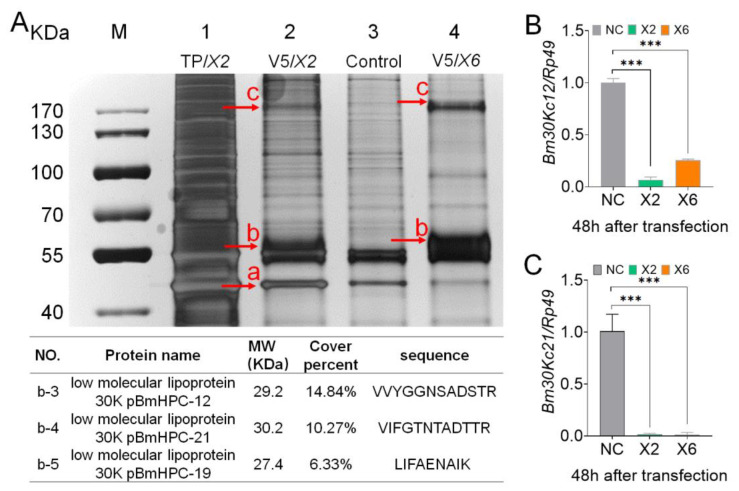
Screening of BmFHL2-interacting proteins. (**A**) 30K proteins were obtained by silver staining and mass spectrometry analysis of the Co-IP products. M: marker; 1: overexpression of *BmFhl2-X2* cell total protein; 2: overexpression of *BmFhl2-X2*/V5 mouse anti; 3: PIZT/V5 His/V5 mouse anti; 4: overexpression of *BmFhl2-X6*/V5 mouse anti. Letters a, b, and c indicate differential bands. (**B**,**C**) qPCR analysis of Bm30Kc12 and Bm30Kc21 at 48 h after overexpression. ***, *p* < 0.001. NC: no transfection load. X2: overexpression of *BmFhl2-X2*. X6: overexpression of *BmFhl2-X6*. *Rp49* was used as the qPCR reference gene. Mean ± SD, N = 3.

## Data Availability

The data presented in this study are available in the supplementary material.

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
