# Peer review of "The LIM Domain Protein BmFHL2 Inhibits Egg Production in Female Silkworm, Bombyx mori"

_cells, 2023, doi:10.3390/cells12030452_

Round 1
Reviewer 1 Report
The silkworm, Bombyx mori is an important Lepidoptera model organism and most important industrial insect of the world. The authors studied the function of the FHL2 homologous gene (BmFhl2) of B. mori in egg formation in female pupae by RNAi, overexpression, WB, Co-IP and qPCR. The results are very interesting, that RNAi of BmFhl2 significantly increase eggs (both of produced and laid egg number) by inhibiting the synthesis of 30Kc19. The experiments are well designed. And it is well written in English. But there are still some improvement space.
1. In abstract,there are several "Bombyx mori", except the first one, the other should be abbrivated in short form. But in the "introduction" line 43, the "silkworm" should be limited to B. mori.
2. In line 21 and 27, "protein (family)" was omited after "30K " . Similarly, in line 233, after knockdown....(what).
3.In fig2E, the 3rd picture showed "Rate of lain eggs" should be “Rate of laid eggs”.
4. In fig3B, WB bands were not indicated which is RNAi,which is NC.
5. In fig5B and C, The horizontal axis is showed as "after transfection (h)", but there is no time points.
6. RNAi of BmFhl2 increase the number of eggs significatly, but how about the component composition of eggs, is there any changes? And the hatch ability or healthness of hatched larvae of RNAi group compare to the NC?
Reviewer 2 Report
1.In material section, authors used siRNA for interference. In Fig.2 legend, authors used dsRNA. Is it dsRNA or siRNA?
2.Line99: …equal amount of negative control siRNA (si NC). What’s the si NC or what component of the si NC?
3.30K protein family is composed of multiple genes. After knockdown of BmFhl2, why the authors did not detect the other 30K protein encoding genes, except for 30Kc19.
4.How did the author determine that BmFHL2 is a transcription factor of 30K?
5.The authors speculate that FHL2 may affect egg formation through the insulin signaling pathway. Since there are many research reports on FHL2 homologues in other species, the author can cite more literature in the discussion and introduction to describe the function and potential mechanism of the gene more clearly and thoroughly, which will also help the follow-up research.
Reviewer 3 Report
The aim of the study described in the paper by Yuan et. al. was to evaluate the effect of FHL2 belonging to the FHL ( four-and-a-half LIM domain) protein family on the reproduction of female silkworm Bombyx mori. Because LIM domain is involved in protein–protein interactions and the effect of FHL2 on the reproduction of female insects was not studied so far the subject of the studies seems justified and valuable.
Methods are sufficiently described, and the results clearly presented.
My suggestion concerns using of “Blood” term thorough the whole text. The counterpart of animal blood in insects is haemolymph, so this term should be used.
Line: 296 – endocrine systems instead endocrine hormones
Conclusions should contain some perspectives for the future in the field of FHL2 research and its role in reproduction of insects
Reviewer 4 Report
The Ms entitled “The LIM domain protein BmFHL2 inhibits egg production in the female silkworm, Bombyx mori” by Qian Yuan and co-Authors contributes to the knowledge of the regulation of the female reproduction in a very important model insect, the silkworm Bombyx mori. Authors, by iRNA, overexpression and co-immuno precipitation assays investigated, for the first time in an insect, the role of the FHL2 gene homologue/FHL2 protein involved, in mammals, in many female reproductive processes.
The experiments have been well designed and carried on, the pictures are very explicative (including supplementary materials) and the paper appear clearly written. I highly recommend to publish this Ms in CELLS.
Only a few minor comments:
- Although the variuos experiments have been summarized in the Abstract, I suggest to briefly describe at the end of the Introduction (line 67) the methodology choosen to provide “a theoretical basis for exploring the regulatory mechanism of silkworm egg formation”.
-Abbreviations throughout the text
E20 is only defined in the Abstract section. I suggest to define the name of all proteins/genes beside their abbreviation, the first time in which they appear in the main text (with the exception of the Abstract section) including figure legends, i.e. Rp49 in Figure 1. Note that NC (Figures 2-5), has not been defined (although it could be trivial) in Material and Methods section.
-Figure 2C lacks the measurement of the scale bars.
Add this information directly on the bars or in the figure legend.
-Legend of the Figure S2: ...Bombyx mori.....
Change as italics
-Figure 1B, left picture: fat bady.
Correct as fat body
-In the References 7, 11-16, 18-20, 24-25, 27, 33-34, 36: genus and species name should be in italics
-Lines 77-78…The tissue materials, including gonads, fat bodies, blood, and midgut, were collected…
Authors should explain the mode of remotion of the organs and blood from the silkworm, the type of buffer (if used) and the amount of individual dissected (not only that materials were a mix from three population). Alternatively, they could cite a similar procedure already published, if it is the case.
-Line 169……protein polypetide molecules….
Change as …..polypeptide molecules…..
-Line 181 ……and Trichoplusia ni are members of the Order Lepidoptera which are……
Change as …… and Trichoplusia ni, members of the Order Lepidoptera which are……
-Line 186….prophase? Authors would intend “precocious” stage of pupa? Change accordinlgy.
-Line 188…pre-pupal stage.
Authors should clarify whether the term pre-pupa corresponds to P0 stage (Figure 1 B) or not.
-Line 187….gonads….
The term gonads includes both ovaries and testes. Indeed Authors, here, refer to the ovary, as in the following text they describe the BmFhl2 expression also in testis, blood and midgut. Change accordingly.
-Line 191…. Figure S2…..
Correct as Figure S1 (or S1B)
-Line 199…The comparison species involve four orders of Lepidoptera, Diptera, Rodentia, and Primates, including 14 species in total…
Change as The comparison species involve four orders among Lepidoptera, Diptera, Rodentia, and Primates, including 14 species in total…
-Line 289….in female insects….
Change as ….in female Bombyx mori…
-Line 282….the amount of egg production…..
Change as …. the amount of egg produced….
Line 322… Figure 3C…
Indeed, the text refers to Figure 2C. Change accordingly
